# *Cymbopogon citratus* Essential Oil Increases the Effect of Digluconate Chlorhexidine on Microcosm Biofilms

**DOI:** 10.3390/pathogens11101067

**Published:** 2022-09-20

**Authors:** Luís Felipe Garcia Leal Mouta, Raquel Souza Marques, Cristiane Yumi Koga-Ito, Marcos José Salvador, Elisa Maria Aparecida Giro, Fernanda Lourenção Brighenti

**Affiliations:** 1Department of Morphology, Genetics, Orthodontics and Pediatric Dentistry, Araraquara Dental School, São Paulo State University—UNESP, R Humaitá, 1680, Araraquara 14801-903, Brazil; 2Department of Environment Engineering, Institute of Science and Technology, São José dos Campos, São Paulo State University—UNESP, Av Eng Francisco José Longo, 777, São José dos Campos 12245-000, Brazil; 3Department of Plant Biology, Institute of Biology, University of Campinas—UNICAMP, R Carlos Gomes, 241, Campinas 13083-970, Brazil

**Keywords:** biofilm, essential oils, phytotherapy, dental caries, biological products

## Abstract

The aim of this study was to evaluate the effect of the *Cymbopogon citratus* essential oil and its association with chlorhexidine on cariogenic microcosm biofilm composition and acidogenicity. Minimum inhibitory and bactericide concentrations from the essential oil and chlorhexidine were determined by broth microdilution assay. Microcosms (polymicrobial) biofilms were produced on glass coverslips, using inoculum from human saliva in McBain culture medium (0.5% sucrose exposure for 6 h/day) for 3 days in 24-well plates. The biofilms were treated twice a day and their composition was evaluated by microorganism quantification. The acidogenicity was evaluated by measuring the pH of the spent culture medium in contact with the biofilm. Overall, the association of *C. citratus* and chlorhexidine reduced total bacterial counts and aciduric bacteria (maximum reduction of 3.55 log UFC/mL) in microcosm biofilms. This group also presented the lowest acidogenicity even when exposed to sucrose-containing medium. *C. citratus* essential oil increases the effect of digluconate chlorhexidine on microcosm biofilms. Based on these findings, this study can contribute to the development of new formulations that might allow for the use of mouthwashes for a shorter period, which may reduce undesirable effects and increase patient compliance to the treatment.

## 1. Introduction

Dental caries is a multifactorial disease that affects a great number of the world’s population and is considered as a serious public health issue [1]. The disorganization of dental biofilm by brushing and flossing is a solid strategy for controlling this disease [2,3]. However, complementary approaches with antimicrobial substances have been explored when mechanical biofilm removal is inefficient [4,5].

Chlorhexidine digluconate is an antimicrobial agent with a broad spectrum of action [6]. However, adverse effects were reported, such as tooth pigmentation, oral mucosa irritation [7], taste alteration [8] and increased dental calculus formation [9]. Additionally, studies have pointed out the inability of chlorhexidine to control mature biofilms [10].

*Cymbopogon citratus* (Lemongrass) extracts and essential oils are the target of several studies due to its anti-inflammatory, antifungal, anticancer and antibacterial properties [11,12,13,14,15,16]. Furthermore, it has been widely used in traditional medicine. The *C. citratus* essential oil is composed mainly by antibacterial terpenes, such as neral (cis-citral) and geraniale (trans-citral) [17,18,19]. Recently, studies have shown that the *C. citratus* essential oil has also antimicrobial activity against cariogenic microorganisms in suspension [20,21,22] and in monospecies biofilms [15,18,23,24].

However, there is only one study that evaluated the effect of *C. citratus* using a microcosm (polymicrobial) biofilm [18]. Microbial interaction in biofilms might change its pathogenicity [25]. Models of microcosm biofilms formed from salivary inoculum are used as an alternative to monospecies models, because they represent a version closer to the real oral conditions. As dental caries is a result of metabolic interactions of such diverse communities [26], the relevance of using an experimental microcosm model is to mimic the ecological complexity of the oral microbiota, since the biofilm is grown because of the interaction among different microorganisms;.

A previous study has showed that the association between essential oils and chlorhexidine enhances the antimicrobial effect against biofilms [27]. Thus, the association of of *C. citratus* with chlorhexidine might enhance the limited activity of chlorhexidine in microcosm mature biofilms.

Therefore, the aim of this study was to evaluate the effect of the *C. citratus* essential oil and chlorhexidine digluconate association in microbiological composition and acidogenicity of microcosm biofilms formed in vitro.

## 2. Results

### 2.1. Antibiofilm Activity

The means (±SD) of total bacteria concentration in donor saliva was 5.10 × 10^6^ CFU/mL (±3.30 × 10^5^); for streptococci mutans, it was 3.20 × 10^4^ CFU/mL (±4.71 × 10^3^); for aciduric bacteria, it was 1.96 × 10^5^ CFU/mL (±2.78 × 10^4^).

Both the essential oil and chlorhexidine demonstrated antimicrobial activity against the microcosm inoculum. The MIC and MBC were coincident and the values for *C. citratus* essential oil were 3.12 µL/mL and 0.080 µL/mL for chlorhexidine.

For biofilms, there was a statistically significant difference among all treatments for total aciduric bacteria and total bacteria (*p* ≤ 0.0001). The lowest concentrations of total aciduric bacteria and total bacteria were observed when the association of *C. citratus* essential oil and chlorhexidine was used as treatment. For the streptococci mutans group, there were no statistically significant differences between chlorhexidine and the association of the *C. citratus* essential oil and chlorhexidine treatments (*p* = 0.5797) (Figure 1).

### 2.2. Biofilm Acidogenicity

The association of the *C. citratus* essential oil with chlorhexidine showed the highest pH values on the sucrose-containing culture medium. Only for this association, pH values were above 5.5 (critical pH for tooth enamel demineralization) on days 2 and 3 of the experiment (Figure 2A). On pH variation (∆pH), all groups showed a statistically significant difference for days 2 and 3 (*p* ≤ 0.0001). The lowest values of ∆pH were obtained in biofilms treated with the association of the *C. citratus* essential oil and chlorhexidine from the second day onwards (Figure 2B).

## 3. Discussion

Current studies show that the *C. citratus* essential oil has promising antimicrobial activity [18,20,21,22,23] and low cytotoxicity [18,23]. However, these studies used microorganisms in suspension or monospecies biofilms, raising the question about the performance of this oil in environments closer to the oral cavity.

To date, there is only one study that evaluated the action of the *C. citratus* essential oil on microcosm (polymicrobial) biofilms [18]. However, this study only demonstrated total bacteria reduction in BHI agar. The reduction in *Streptococcus mutans* or acidogenic bacteria was not accessed. Moreover, the ability to inhibit the growth of cariogenic microorganisms it is not the only factor to be explored in an attempt to reduce the development of dental caries [28]. The control of biofilm acidogenicity is an interesting strategy to control dental caries since this virulence factor is essentially associated with tooth decay. The present study brings important information about the ability of the *C. citratus* essential oil alone or in association with chlorhexidine in reducing acidogenic bacteria and the acidogenicity of microcosm biofilms.

Chlorhexidine digluconate is an antiseptic agent with a broad antibacterial spectrum [6,29,30,31]. Its formulations have been widely used in the chemical control of dental biofilm [7]. However, this substance is not able to control mature biofilms [32] and its long-term use favors the occurrence of side effects [6,9]. Therefore, new therapies have been developed to minimize the undesirable effects and to enhance the use of this substance, allowing its prescription for a short period. The results presented by this study demonstrated that the association of *C. citratus* and chlorhexidine digluconate was able to significantly decrease the acidogenicity and microbial viability of biofilms.

Previous studies have shown that it is possible to decrease the concentration of chlorhexidine and maintain the same antimicrobial effect if an essential oil is used in association with this substance [27,33,34]. However, these studies did not consider the action of chlorhexidine in mature biofilms [32,35]. Moreover, the strategy of reducing chlorhexidine concentration to minimize side effects was not based on scientific evidence. To date, there are no data on the relationship between the concentration of chlorhexidine and the incidence of side effects. An additional limitation of these studies is that they used only planktonic cultures or biofilms of a single species. Thus, the effect of the association of essential oils with chlorhexidine on microcosm biofilms remains unknown.

The present study has demonstrated that the association of the *C. citratus* essential oil and chlorhexidine digluconate-based marketed product has enhanced its antibiofilm potential. To the best of our knowledge, there are no data in the literature reporting similar findings using a microcosm biofilm model. In addition, improving the action of chlorhexidine on mature biofilms might reduce its undesirable effects.

Despite the need of clinical studies to confirm the efficacy of and the reduction in the side effects of this association, it is important to highlight that the biofilm growth model used in this study is based on a simple and effective method to study the effect of treatments against dental caries with important characteristics for the developing of oral biofilms [36].

First, the microcosm biofilm was originated from natural microflora, which has a diverse and complex microbiota, closer to that found in oral biofilm when compared with monospecies biofilms. This allows the study of biofilm control treatments from a microbiological and ecological perspective [37]. Moreover, the biofilms grew on an active attachment model that promotes the formation of a biofilm only with cells capable of adhering to specimens, avoiding microorganism adherence by gravity [38].

The intermittent sucrose exposure regime used on the study simulates the nutrient availability in the oral cavity, and the pH cycles that occur because of intermittent exposure to fermentable carbohydrates, creating a biofilm exposed to conditions close to those found in vivo. These factors may provide more clinically relevant results than biofilms of single species or with biofilms of defined species studies. This is the first study that used biofilms with the combination of these three characteristics to test the association of essential oils and chlorhexidine.

Although the concentration of streptococci mutans in the biofilms treated with the association of the *C. citratus* essential oil and chlorhexidine has remained the same as chlorhexidine-treated biofilms, it was significantly different from the control. Additionally, a significant reduction in the total bacteria and aciduric bacteria count was observed. From an ecological point of view, the Extended Theory of Caries proposed by Takahashi and Nyvad [39] suggests that the control of these groups, particularly the aciduric bacteria, is more important than the reduction in specific species. This mechanism will maintain oral biofilm homeostasis, preventing the emergence of the streptococci mutans group and other species related to caries progression.

The significant reduction in microorganisms in the group treated with the association of chlorhexidine and the *C. citratus* essential oil can be explained by a synergism between major compounds of the essential oil of *C. citratus* (citral and geranial [18]) and chlorhexidine digluconate. Some studies demonstrate a significant reduction in the activity of glucosyltransferases in biofilms treated with citral and geranial substances. These enzymes are responsible for the synthesis and secretion of glucans (main constituents of the Extracellular polysaccharides matrix) [40]. Kumari et al. [41] also observed damage to the surface of biofilms treated with citral. Thus, the degradation of the EPS matrix can make microorganisms in the biofilm more susceptible to antimicrobials, such as chlorhexidine. However, the major components have not been tested in isolation because studies in the literature demonstrate that the interaction between various components of the oils can acquire antimicrobial effects that are not obtained when tested isolated [42,43].

In addition, the present study also showed a significant reduction in the acidogenicity of biofilms. The association of chlorhexidine and the essential oil was the only one able to decrease pH variation in the presence of sucrose. Moreover, the group was also the only one to maintain the pH above that 5.5 from the second day of the experiment (Figure 2). These findings are important since below this pH the hydroxyapatite present in tooth enamel starts to dissolve, which might lead to dental caries [44]. The ability of other plant extracts to control biofilm acidogenicity was previously reported in the literature [45].

Despite the encouraging results, the findings of this study have some limitations, such as the choice of the substratum for biofilm growth (glass coverslips) and the absence of molecular identification of bacterial species. The use of glass coverslips is widely used [27,38,46,47,48] and the literature acknowledges its significance and contribution for biofilm experimental studies [49], since it represents a feasible and low-cost alternative for initial studies [48]. Other methodology substrates should be used in the future to reassure the findings of this study.

Although molecular analysis allows for the identification of a higher number of microbial species/taxa than culture methods, our choice for the culture methods was based on the following: (i) the diversity identified by molecular methods does not correlate with diversity of function or the amount of viable bacteria [50] (ii) some species may not be detected by molecular methods [50,51] and (iii) culture methods improve the detection of *S. mutans* and other aciduric species [52]. Therefore, further studies should be conducted in order to gain a better understanding of the mechanisms of action of this association through the use of different methods.

## 4. Materials and Methods

### 4.1. Ethical Aspects

The use of saliva for biofilm growth was previously approved by the Araraquara School of Dentistry-UNESP Ethics Committee, Protocol No. 52932716.8.0000.5416 (Araraquara, SP, Brazil). After the consent form was signed, stimulated saliva using an unflavored gum base (Parafilms) was collected from a healthy [53] male volunteer aged 26 years, who had not been under antibiotic therapy for at least 6 months. Saliva was collected in the morning and the donor abstained from oral hygiene for 24 h and fasted for at least 2 h before the collection. Saliva was collected in sterile tubes for 5 min. Fresh stimulated saliva from the same donor was used throughout the experiment.

After collection, initial microbial concentration (CFU/mL) of total bacteria [38], streptococci mutans [54] and aciduric bacteria [55] was assessed. Total bacteria were assessed in Wilkins-Chalgren agar; Streptococci mutans was assessed on Mitis Salivarius agar supplemented with 15% sucrose and 0.2 IU/mL bacitracin (MSBS) and aciduric bacteria were assessed on BHI pH 4.7 agar. Incubation was carried out at 37 °C and 5% CO_2_ for 48 h (Wilkins-Chalgren and MSBS) or 96 h (BHI pH 4.7). Next, the number of Colony-Forming Units (CFU) was obtained, and results were expressed in log (CFU/mL).

### 4.2. Minimum Inhibitory Concentration (MIC) and Minimum Bactericidal Concentration (MBC)

The Minimum Inhibitory Concentration (MIC) and Minimum Bactericidal Concentration (MBC) of the Cymbopogon citratus essential oil (Laszlo Aromatherapy Ltd.a, Belo Horizonte, MG, Brazil—batch 0717/05209/F) were carried out using the broth microdilution technique based on the Clinical Laboratory Standards Institute [56] modified by Brighenti et al., 2014 [57].

The essential oil was diluted in Tween 80/McBain broth [58] in a ratio of 1:1:8 (*v/v/v*). McBain broth is composed by porcine gastric mucin (2.5 g/L), bacteriological peptone (2.0 g/L), tryptone (2.0 g/L), yeast extract (1.0 g/L) NaCl, (0.35 g/L), KCl (0.2 g/L), CaCl_2_ (0.2 g/L), cysteine hydrochloride (0.1 g/L), hemin, (0.001 g/L) and vitamin K1 (0.0002 g/L), pH 7.0. The essential oil diluent (Tween 80/McBain culture medium in 1:1:8 (*v/v/v*)) was used as control group. A marketed product containing 0.12% chlorhexidine (“CHX”; Periogard^®^; Colgate-Palmolive industrial Ltd, São Bernardo do Campo, São Paulo, Brazil) was used for comparative purposes.

Briefly, 100 μL of the diluted essential oil or 50 μL of 0.12% chlorhexidine digluconate and 50 μL of twice concentrated McBain broth supplemented with 1% sucrose were added to the first column of 96-well plates. Two-fold serial dilutions were prepared in McBain broth supplemented with 1% sucrose. Concentrations between 0.1 μL/mL and 100 μL/mL (*v/v*) of the essential oil and between 0.0006 μL/mL and 0.6 μL/mL (*v/v*) of CHX were obtained.

A total of 20 μL of fresh saliva was added to each well and the plates were incubated for 24 h at 37 °C, 5% of CO_2_ [36]. After that, the content of each well was sub-cultured in BHI agar to evaluate bacterial growth [57]. MIC was determined as the lowest concentration capable of inhibiting growth and CBM was determined as the lowest concentration capable of resulting in microbial death. The experiments were performed in duplicate.

### 4.3. Mature Microcosm Biofilm Growth Conditions and Treatments

The microcosm biofilm model described by van de Sande et al. [59] and modified by Albuquerque et al. [36] with an intermittent sucrose exposure (6 h/day) was used [36]. Biofilms were grown on glass coverslips (ø 13 mm; n = 12/group) vertically positioned to create an active adhesion model [38] modified by [48]. The coverslips were transferred to 24-well plates (Techno Plastic Products AG, Labware, Switzerland) containing 0.4 mL of fresh saliva and 1.8 mL of McBain broth with 0.5% sucrose [36]. After incubation at 37 °C, 5% of CO_2_ for 6 h, the coverslips containing the biofilms were washed in saline solution 0.9% and immersed in 1.8 mL of the treatment solution for 1 min.

The following treatment solutions were used: (a) *C. citratus* essential oil 10 × MIC; (b) 0.12% chlorhexidine digluconate; (c) *C. citratus* essential oil 10 × MIC and 0.12% chlorhexidine digluconate association; and (d) McBain medium without sucrose (negative control). The concentration of 10 × MIC was chosen based on the literature [18] showing that lower concentrations had no significant effect on biofilms.

After the treatment, the coverslips were washed in saline solution 0.9% again and were transferred to McBain medium without sucrose for an additional 18 h, completing the first 24 h of the experiment. For the next 48 h, treatments were performed twice a day as previously described.

After 72 h, the microbiological composition of the biofilms was analyzed. The biofilms formed on the coverslips were sonicated (Cristofoli ultrasonic bath, Campo Mourao, PR, Brazil, 42 kHz) for 10 s in 2 mL of 0.9% NaCl [36]. Samples were serially diluted in sterile saline and seeded on blood agar [38], MSBS agar [54], and BHI agar (pH 4.7) [55], for the quantification of total bacteria, streptococci mutans and aciduric bacteria, respectively, as previously described. The colonies of growing microorganisms were counted, and the results were expressed as Log CFU/mL.

The biofilm acidogenicity was analyzed by measuring the pH of spent culture medium using an electrode coupled to an ion analyzer (Quimis pH meter, Diadema-SP, Brazil). The pH variation (ΔpH) for each treatment on each day analyzed was calculated according to the following equation: ΔpH = pH (without sucrose) − pH (with sucrose).

### 4.4. Statistical Analyses

The data obtained in the MIC and MBC tests were analyzed descriptively. For the biofilm microbiological analysis, the data distribution was considered normal (Shapiro-Wilk test) and there was variance homogeneity. Thus, ANOVA followed by Tukey’s test was used. For ∆pH, Welch’s ANOVA and Games-Howell tests were used. For all analyses, the significance level set was 5%.

## 5. Conclusions

The present study shows promising results of the *C. citratus* essential oil associated with chlorhexidine for treating polymicrobial biofilms. Overall, this study demonstrated that the association of *C. citratus* essential oil with chlorhexidine digluconate controls pathogenic composition and decreases the acidogenicity of polymicrobial biofilms. These findings might allow for the use of mouthwashes for a shorter period, which may reduce undesirable effects and increase patient compliance to the treatment. Further studies might be necessary to elucidate the mechanism of action of this association and tests in different methodologies.

## 6. Patents

The authors hold a patent request for a product used in the study, by the National Institute of Industrial Property—INPI/SP, on 1 April 2019 under number BR10201900657. All authors approved the publishing of the manuscript.

## Figures and Tables

**Figure 1 pathogens-11-01067-f001:**
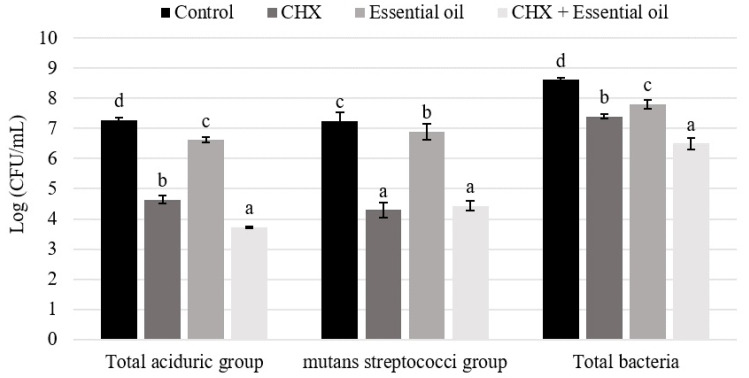
Means ± SD of the concentration (Log CFU/mL) of total aciduric group, streptococci mutans group and total bacteria present in the biofilm. Different letters show significant differences among the groups (Tukey, *p* < 0.05).

**Figure 2 pathogens-11-01067-f002:**
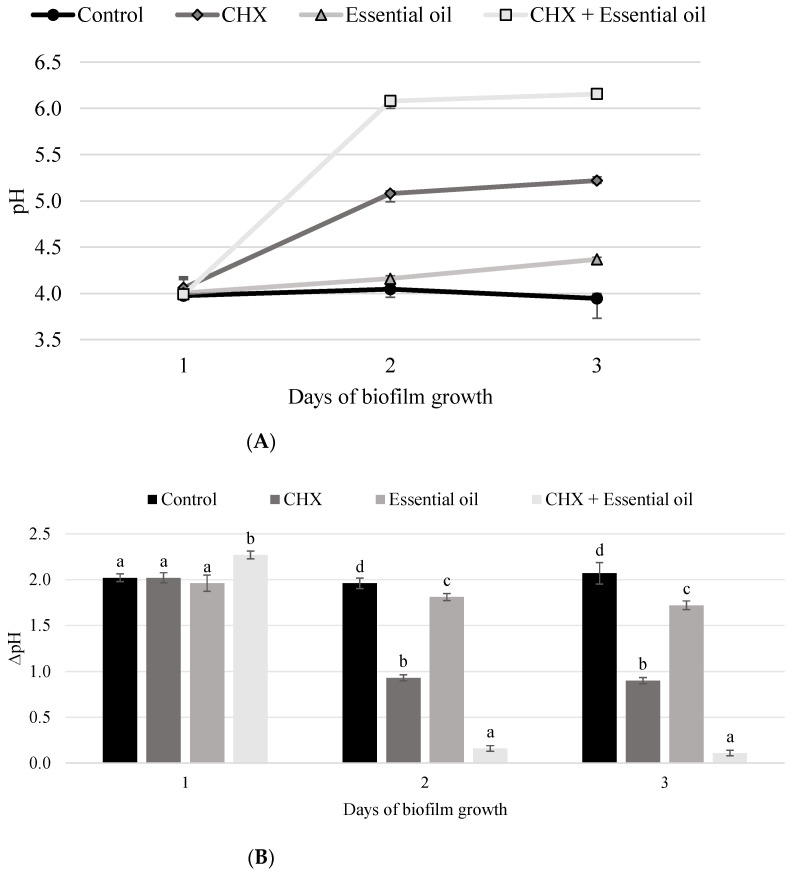
(**A**) pH profile of the groups throughout the experiment. (**B**) Means ± SD of pH variation (∆pH) for the different groups through the experiment. Different letters (a, b, c, d) show significant differences among the groups for each day (Games-Howell, *p* ≤ 0.0001).

## Data Availability

Not applicable.

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
