# Peer review of "Cymbopogon citratus* Essential Oil Increases the Effect of Digluconate Chlorhexidine on Microcosm Biofilms"

_pathogens, 2022, doi:10.3390/pathogens11101067_

Round 1

Reviewer 1 Report

The manuscript is quite written well. However, there are major issues that should be clarified or corrected by the authors:

Materials and Methods: this section is very weak. Authors only used simple 96-wells plates to evaluate the antibiofilm efficacy. The mechanism of destruction of biofilm was not performed. For evaluating antibiofilm assay, CLSM (dual stanning) technique, SEM and TEM are recommended to know the mechanism and effects of any compounds on biofilm thickness.   The reviewer rejects the manuscript for publication because of lack of methodology.

Reviewer 2 Report

1. Regarding the abstract part, it is well designed and written, but the Introduction part needs some improvement as in references 22 and 23 there is error in citation also add in this section the major Cymbopogon citratus essential oil chemical composition also correlate with these chemical constituents in the antimicrobial activity in the discussion part.

2. In the material and methods, a statistical analysis subsection is missing.

3. The conclusion part needs major improvements to contain your study's main outcomes, your own conclusions and recommendations and your future plan.

4. The authors carefully must polish the grammar, typos, and editing throughout the whole manuscript which is mandatory to be done by a native speaker.

5. If possible replace out-of-the-date references by adding more updated ones.

6. All the Latin names of plant or microbial strains must be in the Italic manner

Reviewer 3 Report

Thanks for the opportunity to review this research. The manuscript entitled „ Cymbopogon citratus essential oil increases the effect of digluconate chlorhexidine on microcosm biofilms” have described the effect of Cymbopogon citratus essential oil and the association with chlorhexidine on cariogenic microcosm biofilms composition and acidogenicity. The subject of the manuscript is topical, but I recommend the publishing of the paper after the necessary corrections.

1.     There are several typographical mistakes as well in whole manuscript. Therefore, the author’s thoroughly careful check the language and typo mistake to minimize the error.

2.     Check and format the citations in the whole manuscript. Also, Appropriate references must be provided to explained the background, what is already done and why this study carried out. Hypothesis statement is missing in the introduction section.

3.     The discussion provided by authors is difficult to follow and verify due missing critical details in the methodology section.

4.     The manuscript conclusion must be substantially rewritten.

Round 2

Reviewer 1 Report

If the author agrees to publish as communication, then it's fine. Accepted from my side.

Reviewer 2 Report

The authors did all the required corrections